# Molecular Analysis of Dihydropteroate Synthase Gene Mutations in *Pneumocystis jirovecii* Isolates among Bulgarian Patients with *Pneumocystis* Pneumonia

**DOI:** 10.3390/ijms242316927

**Published:** 2023-11-29

**Authors:** Nina Tsvetkova, Rumen Harizanov, Iskra Rainova, Aleksandra Ivanova, Nina Yancheva-Petrova

**Affiliations:** 1Department of Parasitology and Tropical Medicine, National Centre of Infectious and Parasitic Diseases, 26 Yanko Sakazov Blvd., 1504 Sofia, Bulgaria; tsvetkova@ncipd.org (N.T.); rainova@ncipd.org (I.R.); aleksandra.ivanova@ncipd.org (A.I.); 2Department for AIDS, Specialized Hospital for Active Treatment of Infectious and Parasitic Diseases, Ivan Geshev Blvd. 17, 1431 Sofia, Bulgaria; dr.yahcheva@abv.bg

**Keywords:** *Pneumocystis* pneumonia, dihydropteroate synthase gene, mutation, genotyping

## Abstract

*Pneumocystis jirovecii* pneumonia (PCP) is a significant cause of morbidity and mortality in immunocompromised people. The widespread use of trimethoprim-sulfamethoxazole (TMP-SMZ) for the treatment and prophylaxis of opportunistic infections (including PCP) has led to an increased selection of TMP-SMZ-resistant microorganisms. Sulfa/sulfone resistance has been demonstrated to result from specific point mutations in the *DHPS* gene. This study aims to investigate the presence of *DHPS* gene mutations among *P. jirovecii* isolates from Bulgarian patients with PCP. A total of 326 patients were examined via real-time PCR targeting the *P. jirovecii* mitochondrial large subunit *rRNA* gene and further at the *DHPS* locus. *P. jirovecii* DNA was detected in 50 (15.34%) specimens. A 370 bp *DHPS* locus fragment was successfully amplified in 21 samples from 19 PCP-positive patients, which was then purified, sequenced, and used for phylogenetic analysis. Based on the sequencing analysis, all (*n* = 21) *P. jirovecii* isolates showed *DHPS* genotype 1 (the wild type, with the nucleotide sequence ACA CGG CCT at codons 55, 56, and 57, respectively). In conclusion, infections caused by *P. jirovecii* mutants potentially resistant to sulfonamides are still rare events in Bulgaria. *DHPS* genotype 1 at codons 55 and 57 is the predominant *P. jirovecii* strain in the country.

## 1. Introduction

*Pneumocystis jirovecii*, a fungus formerly known as *Pneumocystic carinii*, can cause a lung infection mainly associated with pneumonia in patients with acquired immunodeficiency syndrome (AIDS) [1], patients who are immunocompromised but not infected with human immunodeficiency virus (HIV) (e.g., individuals who underwent solid organ transplantation [2], patients with acute leukaemia and other haematological malignancy [3], and those treated with immunosuppressive agents including steroids [4]), as well as malnourished persons [5,6], infants [7,8], and young children [9,10,11].

*Pneumocystis* pneumonia (PCP) is a significant cause of morbidity and mortality in patients with AIDS [12,13], although the use of combined antiretroviral therapy (cART) has contributed to a significant decline in the PCP incidence in this population worldwide since the 1990s [14]. A more unfavourable outcome from *Pneumocystis jirovecii* infection was observed in non-HIV-infected patients compared to HIV-infected patients [15]. A high mortality rate was registered in non-HIV-infected patients with PCP who had inflammatory diseases and were treated with glucocorticoids [16].

The primary option for the treatment and prophylaxis of PCP is cotrimoxazole (trimethoprim-sulfamethoxazole, TMP-SMX), a drug combination that targets two enzymes in the folate pathway—dihydropteroate synthase (DHPS) and dihydrofolate reductase (DHFR). The effectiveness of this combination for PCP treatment has been demonstrated over the years [17,18,19,20]. PCP incidence has declined as a result of prophylaxis and effective antiretroviral therapy [14]. Prophylaxis dramatically lowers the risk for development of PCP in susceptible populations. In patients at high risk of reinfection or recurrence of a previous infection due to incompletely eradicated *P. jirovecii*, the possibility of prophylaxis should be considered [13,21,22]. HIV-infected patients who have never had a PCP episode when their CD4+ cell count is less than 200/mm^3^ are also indicated for PCP prophylaxis [13].

Dihydropteroate synthase (DHPS) is an enzyme that catalyses the condensation of 6-hydroxymethyl-7,8-dihydropteridine pyrophosphate with para-aminobenzoic acid (*p*ABA) to form 7,8-dihydropteroate (DHP) and pyrophosphate [23], a key step in the biosynthesis of folic acid [24]. This enzyme is essential for the de novo synthesis of folates in lower eukaryotes (protozoa and yeast). Folates are essential nutrients (cofactors) involved in the synthesis of nucleotides and the metabolism of amino acids in all living organisms [25,26,27]. Sulphonamide and sulfone compounds are effective inhibitors of DHPS as their structure is similar to *p*ABA. They act as substrates for DHPS that lead to enzyme inhibition due to direct competition with *p*ABA [17,28,29,30]. Mutations in the DHPS gene cause resistance to sulphonamides [31,32,33,34,35,36].

Opportunistic infections including PCP are still a major public health problem in Bulgaria among patients living with HIV and AIDS. At risk are also non-HIV-infected patients who are otherwise immunocompromised, including sufferers of chronic lung diseases, cancer, inflammatory or autoimmune diseases, and those who underwent solid organ transplants or stem cell transplantation. According to the Country Health Profile for 2020 [37], the number of individuals diagnosed with cancer was significant. The new cancer cases for 2020 in Bulgaria were around 35,000, and as reported by the European Cancer Information System (ECIS) of the Joint Research Centre, during the past 10 years, there has been an increasing trend in mortality [38]. The information from the Bulgarian National Primary Immune Deficiencies (PID) Registry shows that the number of newly diagnosed patients with PID has more than doubled (177%, representing a rate of 2.7 per 100,000) since 2014 [39]. In a study on the distribution of *Pneumocystis* infection in infancy in Bulgaria, which was conducted according to clinical and epidemiological indications between 1976 and 1979, 123 children aged from 16 days to 10 months were examined. In 40 (32.52%) of them, cysts of the pathogen were identified via microscopic examination of Giemsa-stained smears from respiratory tract secretions [40]. With the detection of this pathogenic fungus in AIDS patients in the 1990s, the interest of Bulgarian researchers in *Pneumocystis* rose [41]. *Pneumocystis* pneumonia was reported as the second most frequent opportunistic infection detected in HIV-positive patients in the country. Based on microscopic examination of stained smears (e.g., stained with Giemsa, toluidine blue, and methenamine-silver nitrate according to Gomori) for direct detection of *Pneumocystis*, between 6.06% and 8.2% of suspected patients (*n* = 165 and *n* = 170, respectively) were diagnosed with pneumocystosis, as reported in studies by Kurdova et al. (2004) and Georgieva et al. (2012) [42,43]. With the implementation of real-time Polymerase Chain Reaction (PCR) in the laboratory diagnostic practice, detection of *P. jirovecii* was significantly improved, which led to an increase in the frequency of pathogen detection in the examined clinical specimens at the National Reference Laboratory for Diagnosis of Parasitic Diseases, Department of Parasitology and Tropical Medicine, National Centre of Infectious and Parasitic Diseases. Ivanova et al. (2019) reported a significantly high rate of positive results in the HIV-infected patient group (8+, 72.7%, *n* = 11) compared to the HIV-negative group (3+, 13.6%, *n* = 22) of examined patients for the period September 2017–April 2019 [44].

The objective of this study was to investigate the presence of *DHPS* gene mutations among isolates of *P. jirovecii* from HIV-positive and non-HIV-infected patients with PCP in Bulgaria.

## 2. Results

PCP diagnosis was confirmed by the detection of *P. jirovecii* DNA in 50 (15.34%) cases out of the total 326 (68 HIV-infected, 27 non-HIV immunocompromised, and 231 non-immunosuppressed) patients examined using real-time PCR targeting the mtLSU rRNA gene of *P. jirovecii*. Table 1 shows the demographic and clinical data for the 50 patients whose respiratory specimens were positive for *P. jirovecii* DNA. Of the 50 patients, 41 were males and 9 were females (Table 1). The median age was 35 years (range, 0.4–61 years). Most of the positive specimens (84%, *n* = 42) were obtained from immunocompromised individuals, including patients with HIV (*n* = 36) and those receiving medications that suppress immune function (*n* = 6). In the group of patients without data for immunosuppression, eight specimens were positive according to real-time PCR (Table 1). None of the HIV-negative patients (*n* = 14) had received a sulfa/sulfone-containing agent as a therapy before the episode of PCP.

A 370 bp fragment of *P. jirovecii DHPS* locus was successfully amplified in 21 specimens (designated as BG1 to BG19) obtained from 19 PCP-positive patients (Patient IDs P1 to P19). Table 2 presents the details of patients studied for *DHPS* gene mutations. In two of the HIV-infected patients (P8 and P18), two control specimens were tested because their complaints of shortness of breath, cough, and ongoing fever had continued despite the courses of treatment with TMP/SMZ (Table 2).

The alignment of the obtained sequences was compared with the published *DHPS* wild-type sequences of *P. jirovecii* (GenBank accession numbers AY628435, AF139132, U66282, U66280, and AJ586567 [45,46,47,48]) and with the *DHPS* mutant-type sequences (GenBank accession numbers U66278 and U66281 [47]) of *P. jirovecii* (Figure 1).

The sequencing analysis showed that all of the 21 *P. jirovecii* isolates were of *DHPS* genotype 1 (the wild type with the following nucleotide sequence ACA CGG CCT at codons 55, 56, and 57, corresponding to amino acids threonine (at codon 55) and proline (at codon 57), respectively). The sequences of isolates BG8-2 and BG18-2 from the control specimens of patients P8 and P18 were identical to the sequences obtained from the first specimens (BG8-1 and BG-18-1) of the patients.

Based on the quantitative PCR results and clinical manifestation (intermittent subfebrility and nonproductive cough) at day 123 (D + 123) (the counting started on the day when the first sample was examined for the presence of *P. jirovecii* DNA − D + 1), PCP recurrence was assumed in patient 18 (P18) [49]. The sputum specimen obtained (isolate BG18-2) revealed an increase in the copy number of mtLSU rRNA *P. jirovecii* gene/mL to the level detected in the first sample tested (D + 1). The sequencing analysis for mutations at codons 55 and 57 of this isolate showed wild genotype 1. Despite the quite prolonged detection of the pathogen by means of real-time PCR targeting the mtLSU rRNA gene of *P. jirovecii*, patient therapy was successful, no *P. jirovecii* DNA was detected in the control sputum specimen one year after the last negative sample (December 2020, D + 310), and the patient had no clinical complaints from the respiratory tract as well to the designated end of this study.

PCP was the cause of death in six of the HIV-infected patients enrolled in the current study (Table 2), despite no mutations at codons 55 and 57 being detected in the isolates from their sputum specimens.

Phylogenetic relationships

A phylogenetic tree was generated based on the sequencing analysis of the partial *DHPS* gene of *P. jirovecii* using the MEGA program version 11.0.13 (Figure 2). The phylogenetic analysis of 21 *P. jirovecii* isolates (in the case of patients P8 and P18, the isolates obtained from the control specimens were used as well) showed that the isolates were dispersed in two main clades. The first main clade included all of the Bulgarian isolates and all (except U66280) of the previously published *DHPS* wild-type sequences used as reference. The second clade consisted of two sequences used as representatives of the *DHPS* wild type (accession numbers U66278 and U66281 [47]).

## 3. Discussion

At the beginning of the acquired immunodeficiency syndrome (AIDS) epidemic, PCP was recognised as a common life-threatening (AIDS-indicator) opportunistic infection. It occurred predominantly in patients who developed AIDS [50], as well as among individuals who were unaware of their HIV-positive status or in those who did not adhere to the therapy prescribed [51]. The first report of *Pneumocystis* pneumonia associated with the starting of the HIV epidemic was published in 1981 and was related to four homosexual men from Los Angeles, who were diagnosed with AIDS [52]. Between 1979 and 1981, an outbreak of community-acquired *Pneumocystis* pneumonia among 11 young male drug abusers and gay men from New York City was described [53]. The widespread use of cART and primary PCP prophylaxis contributed to the decrease in PCP cases among the HIV-infected population. A low incidence of primary PCP in patients receiving cART who had virologically suppressed HIV infection, with CD4 cell counts < 200 cells/mL, irrespective of prophylaxis, was suggested based on data presented in an observational HIV epidemiological research in Europe among patients with AIDS [54]. A multicohort analysis of HIV-infected persons in the United States and Canada revealed an incidence of PCP of <1 case per 100 person-years [55].

In the period 2019–2021, a decrease in new HIV diagnoses reported from countries of the EU/EEA was observed, with 24,801, 14,971, and 16,624 individuals and rates of 5.4, 3.7, and 4.3 cases per 100,000, respectively. For the same period, the reported AIDS cases from EU/EEA countries were as follows: 2772 in 2019, 1760 in 2020, and 1895 in 2021—a crude rate of 0.5 cases per 100,000 population [56,57,58].

In Bulgaria, since the beginning of the HIV/AIDS epidemic (1986) until 2021, 3721 individuals were diagnosed with HIV [59,60]. A total of 1950 persons were hospitalised for treatment of HIV infection or were subsequently followed up in 2021. Antiretroviral treatment was received by 99% of them, and this resulted in undetectable viral load in 75% of the treated patients [59]. The rates of newly diagnosed HIV in Bulgaria for the years 2019, 2020, and 2021 represented 3.7, 2.9, and 3.4 cases per 100,000 with 258, 299, and 238 cases, respectively. Bulgaria reported 68, 43, and 38 AIDS diagnoses and rates of 1.0, 0.6, and 0.5 cases per 100,000 population for these three years [56,57,58].

Between 2019 and 2021, of all AIDS-indicative diseases, *Pneumocystis* pneumonia was diagnosed most commonly in the EU/EEA countries (21%, 24%, and 23%, respectively), followed by pulmonary and/or extrapulmonary tuberculosis (12%, 12%, and 10%, respectively), wasting syndrome due to HIV (12%, 11%, and 14%, respectively), and oesophageal candidiasis (11%, 12%, and 12%, respectively) [56,57,58].

PCP cases have also been recorded in persons without AIDS and the disease incidence has increased [61,62]. Systemic corticosteroid administration was assumed as a predisposing factor for the development of PCP in 90.5% of patients with histopathological evidence of a primary episode of *Pneumocystis* pneumonia, who developed the disease within a month post-exposure to corticosteroids. The median duration of the therapy was 12 weeks before the development of pneumonia; a total of 25% of these patients had PCP after 8 weeks or less of corticosteroid intake with as little as 16 mg of prednisone [63].

After *Pneumocystis* infection was recognised as an opportunistic infection and a leading cause of morbidity and mortality among the rising population of HIV-infected persons, in 1989, guidelines for prophylaxis against *Pneumocystis* pneumonia were the first HIV-related treatment guidelines published [21]. The need for prophylaxis against PCP is considered in two directions: a primary prophylaxis aiming at the prevention of an initial episode for a person who has never had PCP, and a secondary prophylaxis for prevention of subsequent episodes for a person who has already had at least one episode of PCP [21]. In the growing diverse group of immunocompromised individuals, who are receiving immunosuppressive drugs (including high-dose glucocorticoids) due to solid organ transplantation, or those with cancer on chemotherapy, an increase in the PCP incidence has been noticed, despite the availability of effective prophylaxis [64]. The reported mortality in non-HIV-infected patients was higher than in HIV-infected ones (27% vs. 4%) [61]. In non-HIV-infected immunocompromised persons, treatment with glucocorticoids was considered a risk factor for the development of PCP [63,65,66].

The recommended drug as a first-line prophylaxis (as well as for treatment) against *Pneumocystis* pneumonia is trimethoprim-sulfamethoxazole (TMP-SMZ). In cases of allergic reactions or intolerance to TMP-SMZ, dapsone (a sulfone, which is a DHPS inhibitor) is the alternative prophylaxis choice. TMP and sulfonamides have a broad antibacterial spectrum of activity against different microorganisms, such as gastrointestinal pathogens including bacteria (*Escherichia coli*, *Shigella* species, and *Vibrio* species) and the protozoans *Cystoisospora* and *Cyclospora* species; respiratory tract pathogens (*Streptococcus pneumoniae* and *Haemophilus influenzae*) and the fungus *Pneumocystis jirovecii*; skin pathogen (*Staphylococcus aureus*); and pathogens causing urinary tract infections (*Klebsiella pneumonia* and *Escherichia coli*) and ear infections (*Haemophilus influenzae*) [67,68,69,70]. However, with the widespread use of TMP-SMZ for treatment and prophylaxis of opportunistic infections, reports of an increase in the selection of TMP-SMZ-resistant microorganisms, including *P. jirovecii*, rose [71]. Cases of possible drug resistance in *Pneumocystis* organisms were suspected, and many researchers have studied the reasons for the emergence of such resistance. Animal studies have revealed that the activity of TMP-SMZ against *Pneumocystis* is mainly due to SMZ, which is an inhibitor of the dihydropteroate synthase (DHPS) gene, and not to TMP, an inhibitor of the dihydrofolate reductase (DHFR) gene [72,73,74]. In several microorganisms, sulfa or sulfone resistance has been demonstrated to result from specific point mutations in the *DHPS* gene [31,75]. Many studies have demonstrated point mutations in the *Pneumocystis DHPS* gene and have found an association with the use of sulfa or sulfone drugs for *Pneumocystis* prophylaxis [47,76,77]. In a prospective study, more than half of the newly diagnosed patients with HIV infection harbour a mutant genotype, and of these, 80.3% (57 out of 71) had received sulfa or sulfone prophylaxis against *Pneumocystis* organisms [71].

Most frequently, single nucleotide polymorphisms (SNPs) in the *P. jirovecii DHPS* gene were described as occurring due to two mutations leading to a single nucleotide change. Commonly, they affect the nucleotides at positions 165 (A-G) and 171 (C-T) in a highly conserved region of one of the putative active sites of the enzyme (causing amino acid substitutions at codons 55 (Thr to Ala) and 57 (Pro to Ser)) [35,77,78] and can be observed as a single or a double mutation in the same *P. jirovecii* isolate). Depending on which nucleotide positions are affected by mutation(s), there are four different *DHPS* genotypes described—wild type in positions 165 and 171 (WW), wild type in position 165 and mutated in position 171 (WM), mutated in position 165 and wild type in position 171 (MW), and mutated in positions 165 and 171 (MM) [35]. Kazanjian and colleagues identified mutations at codons 55 and 57 of *DHPS* in 7 out of 27 patients with PCP [76]. There were observations about the more frequent occurrence of mutations among patients for whom sulpha prophylaxis failed [77,78]. A three-times higher risk of death from pneumonia was reported by Helweg-Larsen et al. (1999) in patients carrying mutant *Pneumocystis DHPS* strains compared to those infected with the wild type-*DHPS Pneumocystis* isolates [79]. In a study from Santiago, Chile, conducted by Ponce et al. (2017), the registered high frequency of *DHPS* mutations was among adult patients with a first episode of PCP who were sulfa drug treatment-*naive* patients and in whom no PCP prophylaxis had been administered. The authors’ assumption for this finding was the possibility of interhuman transmission and selection pressure from sulfa drugs prescribed for other conditions [80].

In our study, no mutations were detected at codons 55 and 57 in all examined isolates. These results indicate a low prevalence of *DHPS* mutations in Bulgarian *P. jirovecii* isolates, which may be explained by the low selective pressure of sulfa drugs on the local strains. Our findings are consistent with studies by other researchers [81,82], who reported 0% mutant prevalence in the French region Brittany and Indian *P. jirovecii* isolates at the positions of codons 55 and 57. A low prevalence of *DHPS* mutations in *P. jirovecii* strains was reported in many studies, e.g., from India [82], Germany [83], and Italy [84]. Although no mutations at codons 55 and 57 were detected during the study of the *DHPS* gene, six patients enrolled in the current study (P9, P10, P11, P12, P13, and P15) had a fatal outcome, with *Pneumocystis* infection as a leading cause of death.

Recently, three novel *DHPS* genetic variations [mutant 96 (21/30; 70%), mutant 98 (14/30; 47%), and double mutant 96/98 (13/30; 43%)] were reported [85]. Patients who developed severe PCP harboured a mutant genotype (either single mutant 96 or mutant 98, or double mutant 96/98). The authors believe that the reason for the fatal outcomes observed (13, 8, and 7 patients of those that harboured mutant 96, mutant 98, and double mutant 96/98, respectively) is that these mutations were induced in the same coding regions of the *P. jirovecii DHPS* gene as those observed at codons 55 and 57 (sites in the enzyme structure known as a “hot spot” region for drug resistance [47]), and they were the cause of similar drug resistance effects. All seven patients infected with the double *DHPS* mutant strain did not respond to the PCP treatment [85]. Another mutant genotype that occurred as a result of novel nucleotide substitution at position 288 (Val96Ile) was identified in patients with severe episodes of PCP, who did not respond to TMP-SMZ treatment and had fatal outcomes [86]. Other point mutations such as those at codons 23, 60, and 111 of the *P. jirovecii DHPS* locus were also suspected to be involved in drug resistance (sulpha/sulphone) [36,47,87,88]. In our study, aside from the most commonly described point mutations at codons 55 and 57, the presence of sequence variations at other studied *DHPS* codons (23, 60, 75, 78, 83, 88, 100, and 111) was examined. No mutations were detected in all Bulgarian *P. jirovecii* isolate sequences. The isolates obtained from patients P8 and P18 during follow-up showed 100% similarity with the first examined isolates of these patients (Figure 1 and Figure 2).

From the literary sources available to us, in Bulgaria, human infections caused by the fungus *Pneumocystis* have been reported as early as 1976 in infant patients. Later studies described pathogen detection in AIDS and HIV-positive patients, recognising *Pneumocystis* pneumonia as the second most frequent opportunistic infection in the country. Although the *DHPS* locus was not successfully amplified from all the *P. jirovecii*-positive isolates and subsequently used for sequencing analysis, the present study is the first carried out in Bulgaria that aimed at investigating the presence of mutations in the *P. jirovecii DHPS* gene commonly associated with sulfonamide resistance. The accumulated data on the absence of mutations in codons 55 and 57 offer a step forward in our future work to study other parts of the DHPS gene for possible polymorphism(s) in clinical samples obtained from PCP patients from the country. This study is novel for Bulgaria and suggests that *DHPS* mutational analyses should be performed in immunocompromised, infant, and other susceptible patients to avoid treatment failure and fatal outcomes due to PCP.

## 4. Materials and Methods

### 4.1. Study Design

This article is based on data from a prospective study on the prevalence of pneumocystosis among the Bulgarian population, beginning in January 2019.

### 4.2. Ethical Considerations

This study was reviewed and approved by the Institutional Review Board (IRB) 00006384, and informed consent was obtained from the patients. No information that could reveal the identity of the patients who participated in the study was used.

### 4.3. Patients and Samples

During the 4 years of the prospective study (covering the period from January 2019 to December 2022), a total of 326 (68 HIV-infected, 27 non-HIV immunocompromised, and 231 without data for immune suppression) patients were examined for infection with *Pneumocystis jirovecii*. Patients enrolled in this study were with clinical suspicion of *Pneumocystis* pneumonia as they were presented with the following symptoms: fever, dyspnoea, and unproductive cough.

A total of 337 respiratory specimens were collected from 326 patients suspected of having *P. jirovecii* infection, including 195 specimens from the lower respiratory tract (LRT) (61 bronchoalveolar lavage fluids [BALF], 132 sputums, and 2 gastric aspirates), and 142 specimens from the upper respiratory tract (URT) (11 tracheal aspirates and 131 throat secretions).

### 4.4. DNA Extraction and Real-Time PCR for Detection of P. jirovecii

The DNA extraction procedure was performed by using the PureLink™ Genomic DNA Mini Kit (Life Technologies Corporation, Carlsbad, CA, USA) following the manufacturer’s instructions. *P. jirovecii* DNA was detected via real-time PCR assay (RIDA^®^GENE *Pneumocystis jirovecii* real-time PCR Kit, r-biofarm AG, Pfungstadt, Germany) targeting mitochondrial large subunit rRNA (mtLSU rRNA) gene of *P. jirovecii* in the clinical specimens.

### 4.5. Amplification of P. jirovecii DHPS Gene

Clinical specimens tested positive by means of real-time PCR were further studied to detect the *DHPS* gene. Amplifications were carried out with the forward primer AHUM (designated as DHPS-3 in Calderon et al. (2004) [36]) (5′-GCG CCT ACA CAT ATT ATG GCC ATT TTA AAT C-3′) and reverse primer (designated as DHPS-4 in Calderon et al. (2004) [36]) (5′-GGA ACT TTC AAC TTG GCA ACC AC-3′), as described by Lane et al. (1997) [47] and Calderon et al. (2004) [36], and using published touchdown protocol according to Helweg-Larsen et al. (2000) [89]. Amplicons (expected size 370 bp) were run on 2% agarose gels stained with the GelRed^®^ Nucleic Acid Gel Stain, 10000X (Biotium, Inc., Fremont, CA, USA.), and sized using the GeneRuler 100 bp DNA Ladder (Thermo Fisher Scientific Baltics UAB, Vilnius, Lithuania), and fluorescence detection was performed using SYNGENE gel documentation system (GelVue Model No. GVM20, Synoptics Ltd., Beacon House, Nuffield Rd, Cambridge CB4 1TF, United Kingdom).

### 4.6. Sequencing of DHPS Gene and Phylogenetic Analysis

The amplified fragments of DHPS locus via PCR were purified and sequenced by Bioneer Corporation (8-11 Munpyeongseo-ro, Daedeok-gu, Daejeon, Republic of Korea) using previously described forward and reverse primers (Lane et al., 1997 Calderon et al., 2004) [36,47]. Using BioEdit software version 7.2.5 with the Clustal W alignment program, the obtained chromatograms of PCR products were trimmed and aligned. Finally, a 327 bp consensus sequence length was used for phylogenetic analysis. The reference sequences with GenBank accession numbers AY628435, AF139132, U66282, U66280, and AJ586567 [45,46,47,48] for the wild type and with GenBank accession numbers U66278 and U66281 [47] for the mutant type of *P. jirovecii DHPS* locus were used for the analyses. The evolutionary history was inferred using the Neighbour-Joining (NJ) method [90] and the phylogram was constructed in MEGA 11 [91]. Node support was assessed with 100 (1000) bootstrap replicates [92]. The evolutionary distances were computed using the Kimura 2-parameter method [93].

## 5. Conclusions

The present study was conducted in the Department of Parasitology and Tropical Medicine at the National Centre of Infectious and Parasitic Diseases, Sofia, Bulgaria, and is the first that evaluated *DHPS* gene mutations in *P. jirovecii* isolates from Bulgaria. Infections caused by *P. jirovecii* mutants potentially resistant to sulfonamides are still a rare event. *DHPS* genotype 1 (wild type) at codons 55 and 57 is the predominant *P. jirovecii* strain in Bulgaria.

## Figures and Tables

**Figure 1 ijms-24-16927-f001:**
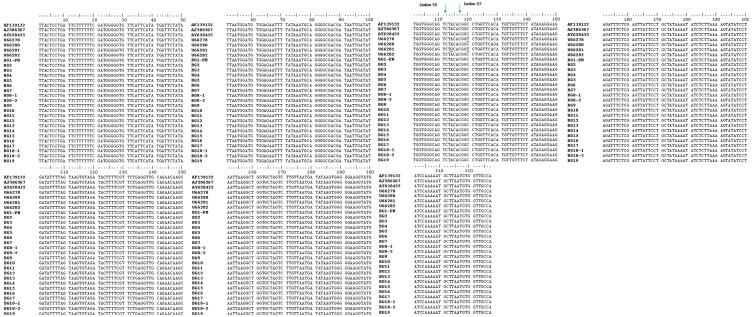
ClustalW Alignment of a 327 bp consensus sequence length of the twenty-one Bulgarian isolates using BioEdit software version 7.2.5. The sequences of the Bulgarian isolates were compared with the previously published *DHPS* wild-type sequences (GenBank accession numbers AY628435, AF139132, U66282, U66280, and AJ586567 [45,46,47,48]) and with the *DHPS* mutant-type sequences (GenBank accession numbers U66278 and U66281 [47]) of *P. jirovecii.* The codons 55 and 57 mutations of the U66278 and U66281 sequences are presented as underlined letters. All sequences of Bulgarian isolates show 100% identity with the published *DHPS* wild-type sequences used as the reference sequences in this work. The GenBank accession number of the deposited sequence of isolate BG1 (BG1-PN) as a representative of all Bulgarian sequences in the current study is OR822222.

**Figure 2 ijms-24-16927-f002:**
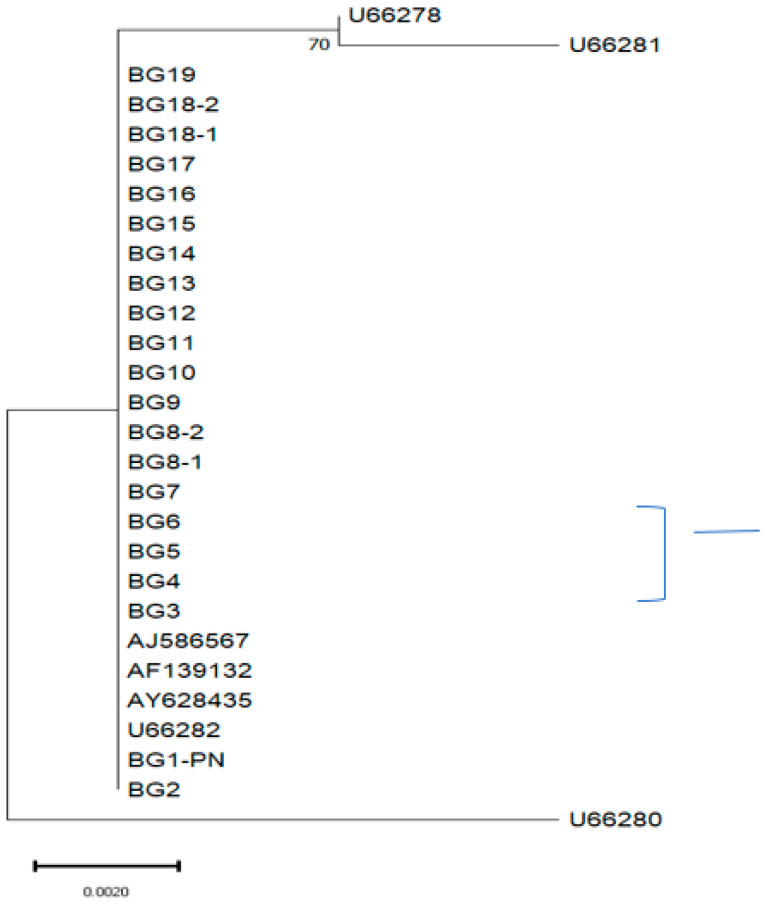
Phylogenetic analysis of partial sequences of the *DHPS* gene from 21 *P. jirovecii* isolates of 19 patients included in this study. The evolutionary history was inferred using the Neighbour-Joining method. The bootstrap consensus tree was inferred from 100 replicates. The percentage of replicate trees in which the associated taxa clustered together in the bootstrap test are shown next to the branches. The tree is drawn to scale, with branch lengths in the same units as those of the evolutionary distances used to infer the phylogenetic tree. The evolutionary distances were computed using the Kimura 2-parameter method and involved 28 nucleotide sequences. Scale bar = 0.0020 substitutions. Evolutionary analysis was conducted in MEGA11. In this study, the sequences present in GenBank with accession numbers AY628435, AF139132, U66282, U66280, and AJ586567 [45,46,47,48] were used as wild-type sequences, while those with accession numbers U66278 and U66281 [47] were used as mutant-type sequences. The GenBank accession number of the Bulgarian isolate BG1 (designated as BG1-PN) deposited sequence as a representative of all Bulgarian sequences analysed in the current study is OR822222.

**Table 1 ijms-24-16927-t001:** Characteristics of patients with positive *P. jirovecii* mtLSU rRNA gene real-time PCR results.

	Immunocompromised (*n* = 42)		Immunocompetent (*n* = 8)	Total = (*n* = 50)
	HIV-Infected	HIV-Negative		
Gender				
Male (%)	30 (71.4)	3 (7.14)	8 (100)	41 (82)
Female (%)	6 (14.3)	3 (7.14)	0	9 (18)
Age in years				
Median (range)	36 (6–54)	35.5 (8–61)	30 (0.4–54)	35 (0.4–61)
Specimens *				50
LRT				47
BAL		1	2	3
SP	36	5	3	44
URT				3
TA			3	3
Underlying conditions ** (No. of patients)				
HIV	36			36
HM		3		3
IPF		2		2
NS		1		1
Pneumonia			7	7
Dyspnoea			1	1

* Specimens: LRT (lower respiratory tract), BALF (bronchoalveolar lavage fluid), SP (sputum), URT (upper respiratory tract), TA (tracheal aspirate). ** Underlying conditions—HIV infection, HM (haematological malignancy), IPF (interstitial pulmonary fibrosis), NS (nephrotic syndrome).

**Table 2 ijms-24-16927-t002:** Data of the patients examined for the presence of *Pneumocystis jirovecii* dihydropteroate synthase gene mutations.

Specimen ID	Patient ID	Age in Years	Sex *	Type of Respirator Specimen #	Underlying Conditions **	Clinical Information	
			F/M			New HIV Diagnosis	PCP Prophylaxis
BG1	P1	0.6	M	TA	pneumonia		No
BG2	P2	15	M	Sputum	pneumonia		No
BG3	P3	14	M	Sputum	HM		No
BG4	P4	8	F	Sputum	NS		No
BG5	P5	60	M	BALF	IPF		No
BG6	P6	44	F	Sputum	HIV/P	Yes	No
BG7	P7	35	M	Sputum	HIV/P	Yes	No
BG8-1 BG8-2	P8	46	M	Sputum	HIV/P	Yes	No
BG09	P09 ^ϯ^	34	M	Sputum	HIV/P	Yes	No
BG10	P10 ^ϯ^	6	M	Sputum	HIV/P	No	Yes
BG11	P11 ^ϯ^	32	F	Sputum	HIV/P	No	Yes
BG12	P12 ^ϯ^	28	M	Sputum	HIV/P	No	Yes
BG13	P13 ^ϯ^	44	M	Sputum	HIV/P	Yes	No
BG14	P14	25	M	Sputum	HIV/P	Yes	No
BG15	P15 ^ϯ^	42	M	Sputum	HIV/P	Yes	No
BG16	P16	35	M	Sputum	HIV/P	Yes	No
BG17	P17	54	M	BLAF	HIV/P	No	Yes
BG18-1 BG18-2	P18	45	M	Sputum	HIV/P	Yes	No
BG19	P19	51	M	Sputum	bronchiectasis		No

* Sex: F (female)/M (male); # Type of respirator specimen—TA (tracheal aspirate), BALF (bronchoalveolar lavage fluid); ** Underlying conditions—HM (haematological malignancy), NS (nephrotic syndrome), IPF (interstitial pulmonary fibrosis), HIV/P (HIV infection with pneumonia); ^ϯ^ a patient with fatal outcome of PCP.

## Data Availability

The datasets generated during the current study are available from the corresponding author upon reasonable request. Because of the full identity between sequences of all Bulgarian isolates included in the current study, one sequence of isolate BG1 (designated BG1-PN) was deposited to the GenBank and it was assigned the accession number OR822222.

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
