# Peer review of "Molecular Analysis of Dihydropteroate Synthase Gene Mutations in Pneumocystis jirovecii Isolates among Bulgarian Patients with Pneumocystis Pneumonia"

_ijms, 2023, doi:10.3390/ijms242316927_

Round 1

Reviewer 1 Report

Comments and Suggestions for Authors

The manuscript from Tsvetkova et al., provides data demonstrating few mutations in the DHPS gene relative to codons associated with enzymatic activity of Pneumocystis jirovecii isolated from Bulgarian patients who were immunosuppressed either because of HIV infection or because of other underlying conditions such as malignancy, interstitial pulmonary fibrosis, or kidney disease.  The authors did phylogenetic analysis and found that the organisms belonged to two different clades based on DHPS sequences.  They concluded that there are few isolates in Bulgaria that are resistant to sulfonamides.  The manuscript provides added evidence that the development of drug resistance by Pneumocystis is slow and in many parts of the world, wildtype organisms predominate.  The data are well presented.  The manuscript is relatively clearly written, though could use an English editor to clean up some sentence structure.

Comments on the Quality of English Language

Some of the phrasing is a bit awkward and could use a native English speaker to proof-read.

Author Response

Dear Reviewer,

On behalf of all the authors of the article, thank you for the positive review.

We revised the text again to improve the style of the sentences.

Reviewer 2 Report

Comments and Suggestions for Authors

Manuscript title:  Molecular analysis of dihydropteroate synthase gene mutations in Pneumocystis jirovecii isolates among Bulgarian patients with Pneumocystis pneumonia

Manuscript ID ijms-2658848

Sequencing analysis of DHPS gene in clinical samples, the mutation of which is associated with sulfa drug resistance, is interesting for comprehensively understanding the mechanism of bacterial drug resistance. However, the present version of manuscript is preliminary, and needs to be carefully revised for publication.

1.     Line 102: PCP diagnosis was confirmed by the detection of P. jirovecii DNA in 50 (15,34%) casesof the total 326… the numbers read confusing here. What is the meaning of the numbers, 15, 34% here? The proportion of number 15 in 50 is 30%, not 34%. This needs to be clarified.

2.     Table 2, what is the meaning of HIV/P? Why is the “HIV” included in HIV/P? This needs to be clarified.

3.     Line 150-154: “The alignment of the obtained sequences was performed in two groups and compared with the published DHPS wild-type sequence, which is part of the folic acid synthesis protein (FAS) gene of P. jirovecii (GeneBank accession no. AY628435). The first group consisted of 25 sequences of isolates BG1 to BG21-1 and isolates BG30 to BG33. Sequences of the nine isolates, BG21-1 to BG21-9 formed the second group.” Where is the figure corresponding to the description of these sentences? The figure should be added.

4.     Figure 1 and Figure2: The DHPS genes from other previously published or reported isolates worldwide should be added in the phylogenetic tree.  Additionally, the red frames in figures should be deleted, unless the authors described the specific aims of these frames.

5.     What are the accession numbers of the sequenced DHPS genes in the isolates of this work?

6.     In the end of discussion, an individual paragraph should be added to discuss the unique finding of this work, and the related importance?

7.     It is suggested that the antimicrobial susceptibility test for the representative isolate should be added to examine the association of the detected mutation and resistance.

Author Response

Dear Reviewer,

We consider your comments to be relevant and will help improve the quality of the article. With respect to individual comments and remarks, our response is as follows:

  1. Line 102: PCP diagnosis was confirmed by the detection of P. jirovecii DNA in 50 (15,34%) cases of the total 326… the numbers read confusing here. What is the meaning of the numbers, 15, 34% here? The proportion of number 15 in 50 is 30%, not 34%. This needs to be clarified.

We apologize for the technical error (comma instead of dot) that misled you. We mean that the proportion of 50 out of 326 is 15.34%. The error has been corrected in the revised text.

  1. Table 2, what is the meaning of HIV/P? Why is the “HIV” included in HIV/P? This needs to be clarified.

We fully agree that we have not properly explained the abbreviation HIV/P in the figure legend. It is about HIV infection with pneumonia. Necessary corrections have been made to the text.

  1. Line 150-154: “The alignment of the obtained sequences was performed in two groups and compared with the published DHPS wild-type sequence, which is part of the folic acid synthesis protein (FAS) gene of P. jirovecii (GeneBank accession no. AY628435). The first group consisted of 25 sequences of isolates BG1 to BG21-1 and isolates BG30 to BG33. Sequences of the nine isolates, BG21-1 to BG21-9 formed the second group.” Where is the figure corresponding to the description of these sentences? The figure should be added.

In the revised article, such figures (Figure 1 and Figure 2)  corresponding to the description of the sentences have been added.

  1. Figure 1 and Figure 2: The DHPS genes from other previously published or reported isolates worldwide should be added in the phylogenetic tree. Additionally, the red frames in figures should be deleted, unless the authors described the specific aims of these frames.

Figures 3 and 4 (the new numbering, after the inclusion of the new two figures) have been adjusted according to your recommendations.

  1. What are the accession numbers of the sequenced DHPS genes in the isolates of this work?

We have not submitted the DHPS sequences obtained from the present study to the genetic sequence database because they do not contain mutations in the studied codons.

  1. In the end of discussion, an individual paragraph should be added to discuss the unique finding of this work, and the related importance?

In the revised text, such a paragraph has been included according to your recommendations.

From the literary sources available to us, in Bulgaria, human infections caused by the fungus Pneumocystis were reported as early as 1976 in infant patients. Later studies described the pathogen detection in AIDS and HIV-positive patients, recognizing Pneumocystis pneumonia as the second most frequent opportunistic infection in the country. Despite the DHPS locus was not successfully amplified from all the P. jirovecii-positive isolates and subsequently used for sequencing analysis, the present study is the first carried out in Bulgaria aimed at investigating the presence of mutations in P. jirovecii DHPS gene commonly associated with the sulfonamide resistance. The accumulated data on the absence of mutations in codons 55 and 57 is a step forward in our future work to study other parts of the DHPS gene for possible polymorphism(s) in clinical samples obtained from PCP patients from the country. This study is novel for Bulgaria and suggests that DHPS mutational analyses should be performed in immunocompromised, infant, and other susceptible patients to avoid treatment failure and fatal outcomes due to PCP.

  1. It is suggested that the antimicrobial susceptibility test for the representative isolate should be added to examine the association of the detected mutation and resistance.

Our goal was to investigate whether, in general, among PCP patients there is a spread of strains that harboured mutant DHPS genotypes because this is the first study in this direction in the country, and not so much to establish a possible connection between the detected mutation and drugs resistance.

Round 2

Reviewer 2 Report

Comments and Suggestions for Authors
  1. What are the accession numbers of the sequenced DHPS genes in the isolates of this work?

We have not submitted the DHPS sequences obtained from the present study to the genetic sequence database because they do not contain mutations in the studied codons.

New comment: submitting and releasing the newly sequenced DNA with access numbers  are necessary for publication. 

Author Response

Dear Reviewer 2, We have followed your recommendation to send the DNA sequences to the Gene Bank, but we still have no answer and cannot guess when in time it will happen.